# Fetal Movement Counting in Prolonged Pregnancies: The COMPTAMAF Prospective Randomized Trial

**DOI:** 10.3390/healthcare10122569

**Published:** 2022-12-18

**Authors:** Louise Moniod, Agathe Hovine, Béatrice Trombert, Florence Rancon, Paul Zufferey, Laura Chauveau, Céline Chauleur, Tiphaine Raia-Barjat

**Affiliations:** 1Department of Gynaecology and Obstetrics, University Hospital of Saint-Etienne, 42000 Saint-Etienne, France; 2Department of Public Health, University Hospital of Saint-Etienne, 42000 Saint Etienne, France; 3INSERM U1059 SAINBIOSE (SAnte, INgénierie, BIOlogie, Saint-Etienne), Jean Monnet University, 42000 Saint-Etienne, France; 4INSERM, Centre d’Investigation Clinique 1408, 42000 Saint-Etienne, France

**Keywords:** fetal movement counting, decreased fetal movements, prolonged pregnancy, neonatal morbidity

## Abstract

In prolonged pregnancies, the risks of neonatal morbidity and mortality are increased. The aim of this trial was to assess the benefits of maternal information about fetal movement (FM) counting on neonatal outcomes in prolonged pregnancy. It was a prospective, single center, randomized, open-label study conducted from October 2019 to March 2022. Intention-to-treat analyses were performed on 278 patients randomized into two 1:1 groups (control group and FM counting group). The primary outcome was a composite score of neonatal morbidity (presence of two of the following items: fetal heart rate abnormality at delivery, Apgar score of <7 at 5 min, umbilical cord arterial pH of <7.20, and acute respiratory distress with mutation in neonatal intensive care unit). There was no significant difference between the two groups in the rate of neonatal morbidity (14.0% in the FM counting group versus 22.9% in the standard information group; *p* = 0.063; OR 0.55, 95% CI 0.29–1.0). In this study, fetal movement counting for women in prolonged pregnancy failed to demonstrate a significant reduction in adverse neonatal outcomes.

## 1. Introduction

Prolonged pregnancies—those progressing beyond 41 weeks of gestation (WG)—represent 17.3% of pregnancies in France, according to the last perinatal survey in 2016 [1]. Beyond term, perinatal mortality increases to 0.2% to 0.3% and perinatal asphyxia increases to 0.6% [2,3,4]. National colleges recommend fetal monitoring two to three times per week after 41 WG [5,6,7].

Decreased fetal movements (FM) may precede an abnormal fetal heart rate (FHR) or in utero fetal death by a few days or weeks [8,9,10], but pregnant women are poorly informed about what to do when they experience it, and some encounter a significant delay in accessing care or do not even contact health care [11,12,13]. Several studies have analyzed the effect of implementing a FM awareness campaign to reduce neonatal morbidity and mortality, but the outcomes are discordant. Some cohort studies have shown a reduction in stillbirth and neonatal morbidity [14,15]. More recent randomized cluster trials did not find differences in neonatal outcomes. Some authors have even shown an increase in the rate of induction, cesarean section, and admission in neonatal units for more than 48 h [16,17,18]. All these studies were performed on pregnant women from 24 or 28 WG. In response to Norman’s study, Walker and Thornton warned against the awareness of fetal movements and suggested limiting such campaigns to full term pregnancies [19].

In prolonged pregnancies, fetal movement monitoring may be an interesting way to improve neonatal outcomes. We conducted a preliminary cohort study that compared a group with no information and a group with information about fetal movement counting. This study showed a reduced number of newborns with more than two of the criteria for a composite score of neonatal morbidity in the FM counting group (10% vs. 24.4%, *p* = 0.0007), with a decrease in the number of consultations for this reason (22.5% vs. 5% *p* = 0.0009) [20].

To our knowledge, no randomized controlled trial evaluating the impact of fetal movement counting in prolonged pregnancies has yet been performed. The aim of the COMPTAMAF study was to evaluate whether fetal movement counting can improve neonatal outcomes in prolonged pregnancies with a prospective randomized trial.

## 2. Materials and Methods

### 2.1. Study Design and Participants

COMPTAMAF was a prospective, single-center, randomized, open-label trial conducted in the Obstetrics and Gynecology Department of the University Hospital of Saint-Etienne, France, between 3 October 2019 and 30 March 2022. Each patient received written information about the study and signed an informed consent form before inclusion. The study was registered at www.ClinicalTrials.gov, accessed on 7 October 2019, under number NCT04117308. Ethical approval was obtained from the ethics committee (Ref 19.05.22.43924).

Each patient entering in the obstetrics department of the Saint-Etienne University Hospital at 41 WG for prolonged pregnancy follow-up was admitted by one of the four midwives in charge of these consultations. During this visit, a midwife asked each eligible patient to participate in the study. In order to avoid travel for pregnant patients, the screening and inclusion visits were conducted during the same consultation. Patients were granted a one-hour reflection time, corresponding to the duration of the consultation. 

The inclusion criteria were: non-pathological mono-fetal pregnancies (except for balanced gestational diabetes), patients affiliated with or entitled to a social security plan, patients aged over 18 years, and patients who had given their participation agreement and signed the consent form. The exclusion criteria were: absence of signature of consent, patient under legal protection or unable to express consent, non-French speaking woman, woman making a maternity change for delivery (risk of follow-up bias), pathological pregnancy (defined as unbalanced diabetes, gravid hypertension, pre-eclampsia, cholestasis, ultrasound abnormalities, and in utero growth retardation), participation in another interventional study, and patient with previous education on FM counting (e.g., patient with previous participation in the before-and-after comparative cohort study with an information brochure on FM counting).

Patients included in the study were randomized into two 1:1 groups. The “control” group received a single standardized information flyer that included a definition of fetal movement and the need to consult if the patient perceived fewer fetal movements, without giving any information on the number of movements that should alert the patient. The “FM counting” group received more detailed written information about fetal movement (definition, interest in monitoring them, technique for counting active fetal movements, chart to be filled out by noting the number of active fetal movements felt at three times of the day, and what to do if the number of active fetal movements decreased). The randomization was centralized using REDCap^®^ online software, hosted at Saint-Etienne University Hospital [21,22], and balanced by blocks of variable size.

All patients included in the study were followed-up with, according to the hospital’s protocol for prolonged pregnancies from 41 WG until delivery, every two days with fetal heart rate monitoring, clinical examination, and questioning. In addition, patients could consult the obstetric emergency unit if they felt FM decreased or for any reason that they considered necessary. Data related to consultations were collected from the patients’ computer records. According to the protocol of our regional perinatal health network, each consultation for decreased fetal movements consisted of fetal heart monitoring, an obstetrical ultrasound with a calculation of the Manning’s score, and a blood test with a Kleihauer’s test. In our study, the consultation was considered as “justified” in the case of an anomaly of the fetal heart rate and/or a Manning score of <8 and/or a positive Kleihauer test.

At the end of each consultation, it could be decided to continue the monitoring at 48 h, to carry out a control visit at 24 h, or to schedule the delivery by induction of labor or by performing a cesarean section. In the context of a consultation for decreased movements, it was decided to induce labor if it was medically justified or to perform a control visit at 24 h. These decisions were made by the obstetrician on call, according to his expertise.

At 41 weeks and 5 days of gestation, if the patient had not delivered, induction of labor was suggested according to the patient’s clinical condition (Bishop score). Thus, the birth was to ideally occur at no more than 42 WG to reduce neonatal risk, which is more substantial after 42 WG. Indeed, the delay between the initiation of labor induction and the delivery could be up to 48 h. Labor, delivery, and neonatal outcome data were collected. All data were collected and managed using REDCap^®^ electronic data capture tools. 

### 2.2. Outcomes

The primary outcome was the number of newborns presenting with at least two neonatal morbidity criteria, according to the score established in our preliminary cohort study [20]:Fetal heart rate at delivery classified as intermediate to pathological according to the 2015 FIGO classificationApgar score of <7 at 5 minUmbilical cord arterial pH of <7.20Acute respiratory distress requiring management in a neonatal intensive care unitCases of in utero fetal demise were collected and considered as an occurrence of the primary outcome

The secondary endpoints were: Related to consultations for decreased FM: number of consultations for this reason, time between the onset of decreased FM and the consultation, classified as less 12 h or 12 h and more according to the timeline identified in the study by Frøen et al. [23], and the number of justified consultationsRelated to changes in obstetrical management: mode of labor onset (spontaneous, induction, or cesarean section before labor) and reasons for induction, and mode of delivery (spontaneous delivery, operative delivery, or cesarean section) and reasons for operative delivery or cesarean section

There were several indications may have led to induction, operative delivery, or cesarean section. Similarly, several methods of induction could be used successively.

### 2.3. Statistical Analysis

In the preliminary cohort study, we found that the number of neonates with 2 neonatal morbidity criteria was greater in the “control” group compared with the “FM counting” group (24.4% vs. 10% *p* = 0.0007) [20]. Assuming a relative decrease of 14% in the number of newborns with 2 criteria of neonatal morbidity in the “FM counting” group, it was necessary to include 278 patients (139 patients in each group) for a power of 90% with an alpha risk of 5%. 

Statistical analyses were performed according to intention-to-treat. Qualitative data were presented in absolute and relative frequencies (in %) and analyzed using the Chi-square test or Fisher’s exact test. Quantitative variables were described by number, mean, and standard deviation. They were compared by Student’s t test in the case of normal distribution and by the Mann–Whitney test in the case of non-normal distribution. For all analyses, a *p* value of <0.05 was considered statistically significant. The odds ratios of the primary endpoint and of each neonatal morbidity criteria were calculated. The influence of demographic and obstetric characteristics, data from consultations for decreased FM, and delivery modalities on the primary outcome were analyzed. All statistical analyses were performed using SAS V9.4 software (SAS Institute Inc., Cary, NC, USA)^®^.

## 3. Results

Between 3 October 2019, and 30 March 2022, 281 patients were randomized, with 141 in the “control” group and 140 in the “FM counting” group (Figure 1). Due to loss of consents, the required number of subjects was not reached, and so we made an amendment to the protocol to include three additional patients. Two patients in the “control” group and one in the “FM counting” group were excluded. The intention-to-treat analyses were performed in 139 patients in each group. Umbilical cord arterial pH could not be performed for eight patients in the “control” group and ten patients in the “FM counting” group because of insufficient quantities of arterial blood samples.

The demographic characteristics of the patients are presented in Table 1. No differences were found between the two groups for maternal and obstetric data.

Table 2 and Figure 2 show the results of the primary outcome. The rate of neonatal morbidity expressed as the composite score was not statistically different between the two groups, but a trend was observed for fewer events in the “FM counting” group (14.0% in the “FM counting” group versus 22.9% in the “control” group (*p* = 0.0630), odds ratio = 0.55 (CI95% 0.29–1.0)). No in utero fetal demise was observed during the study.

Regarding the secondary outcomes, the number of consultations for decreased FM was similar between the two groups (5.8% in the “FM counting” group versus 5.0% in the “control” group), as was the number of justified consultations (Table 3). The time to consultation for this reason was similar in both groups.

Outcomes related to obstetrical management were similar for both groups (Table 4). Five cesarean sections were performed before labor (two in the “FM counting” group and three in the “control” group: two for scarred uterus, two for breech presentation not previously known, and one for contraindication to induction for maternal pathology that did not contraindicate vaginal delivery). The number of labor inductions for decreased FM was similar between the two groups (12.3% in the “FM counting” group versus 9.8% in the “control” group, *p* = 0.67). The gestational age at birth did not differ between the two groups at approximately 290 days, or 41 weeks and 3 days of gestation.

Table 5 shows the findings for the factors influencing the primary outcome. Several variables were associated with neonatal morbidity: primipara (*p* = 0.03), gestational age at delivery (*p* = 0.02), labor induction (*p* = 0.006), and operative delivery and cesarean section (*p* < 0.001). For patients with labor induction, oral misoprostol was related to neonatal morbidity (*p* = 0.04), but the other methods did not seem to have an influence. In the case of instrumental delivery, indications for abnormal FHR induced more neonatal morbidity (*p* = 0.0036), while indications for ineffective expulsive efforts induced less neonatal morbidity (*p* < 0.001).

## 4. Discussion

An information campaign on fetal movement counting in women during prolonged pregnancy was not associated with a reduction in neonatal morbidity. It also did not change the number or timing of visits for decreased FM, nor did it change the obstetrical management of delivery.

Three recent studies have investigated the impact of an information campaign about fetal movements [16,17,18]. These three randomized controlled cluster studies were conducted on 40,000 to 400,000 patients. None of them found a difference in their primary endpoint: stillbirth for two studies and Apgar score of <7 at 5 min for the third. For secondary outcomes, Norman et al. found an increase in the rate of induction, cesarean section, and prolonged (>48 h) duration of admission to the neonatal unit in the group that received the prevention campaign. In contrast, Akselsson et al. found a decrease in the rate of cesarean section and birth gestation above 42 WG and an increase in the rate of the spontaneous start of labor. These studies were conducted in women from the end of the second trimester, with the risk of premature birth caused by decreased fetal movements. Therefore, some authors have suggested limiting information campaigns about fetal movements to specific populations of pregnant women. Walker and Thornton suggested limiting them to women after 37 WG, where neonatal morbidity is reduced by the absence of induced prematurity [19]. Housseine et al. suggest limiting campaigns to patients specifically at high risk of morbidity, particularly those residing in low- and middle-income countries where healthcare accessibility is more difficult and stillbirth rates are higher [24]. In our study, we chose to carry out an information campaign about fetal movement counts in patients with prolonged pregnancies who were at higher risk of neonatal morbidity. We also proposed not to induce labor systematically in the case of decreased FM, but rather, to realize medical exams to limit inductions if medically justified. Thus, we were able to reduce the harmful consequences of neonatal morbidity for non-indicated inductions, but, nevertheless, FM counting did not reduce neonatal morbidity.

In the literature, parity is not related to neonatal morbidity, as found in our study. However, it is associated with an increase in induction, cesarean section, and instrumental delivery, which were factors of neonatal morbidity found in our study that are known in the scientific literature [25]. Oral misoprostol as a method of labor induction was related in our study to neonatal morbidity. This result is not consistent with the Cochrane data on the use of oral misoprostol for labor induction, where there are no increased risks compared to other methods of induction [26]. Further studies on labor induction in prolonged pregnancies may be useful to assess the safety of this method of induction.

The design of our study appeared to be less susceptible to selection bias than previous cluster-randomized studies. Indeed, the randomization in these cluster studies was performed before patient recruitment, whereas in our study, randomization of patients was performed after their inclusion in the study. In addition, the study was offered to all women consulting for follow-up for prolonged pregnancy at 41 WG. However, our study was at greater risk of contamination bias. To limit this, all patients received a written information flyer, whether they were in the “FM counting” group or in the “control” group, and all patients were managed identically, regardless of their group placement.

Although the results showed similar morbidity, there was a tendency for decreased morbidity according to the primary outcome (OR 0.55, 95% CI 0.29–1.04; *p* = 0.063). This can be explained by the lack of statistical power induced by the 18 missing data on arterial pH at the umbilical cord. The missing data were due to insufficient blood samples or laboratory analysis failures. It would, therefore, appear advisable to plan a study with more patients to reach a sufficient power. The neonatal morbidity criteria were determined by a preliminary study conducted in our center [20]. The umbilical cord arterial pH of <7.2 criterium is questionable. Indeed, umbilical cord arterial pH is considered normal if it is >7.24 [27], and the consensus threshold associated with those at high risk of neonatal morbidity and that used in most scientific studies is 7.0. The data from our study could be reproduced using this threshold to be more consistent with the literature, but this will require increasing the number of subjects.

In our study protocol, it was planned to collect the FM charts completed by the patients in the “FM counting” group in order to assess compliance. Very few of these charts were collected (only 11), and it was not possible to determine whether the FM counting was performed for all the patients. However, this allows for a real-life analysis of the results of the FM counting information campaign, whether they performed the count at home or elsewhere.

## 5. Conclusions

In our study, fetal movement counting in prolonged pregnancy failed to demonstrate a significant reduction in adverse neonatal outcomes. It was not associated with a change in the number or timing of consultations for decreased FM and it did not change the delivery outcomes. Nevertheless, our study lacked power, and further studies with more patients appear to be necessary, as well does an analysis of the impact of the FM counting on other populations at risk of neonatal morbidity. It would also be interesting to evaluate the long-term data of children whose mothers had fetal movement count information.

## Figures and Tables

**Figure 1 healthcare-10-02569-f001:**
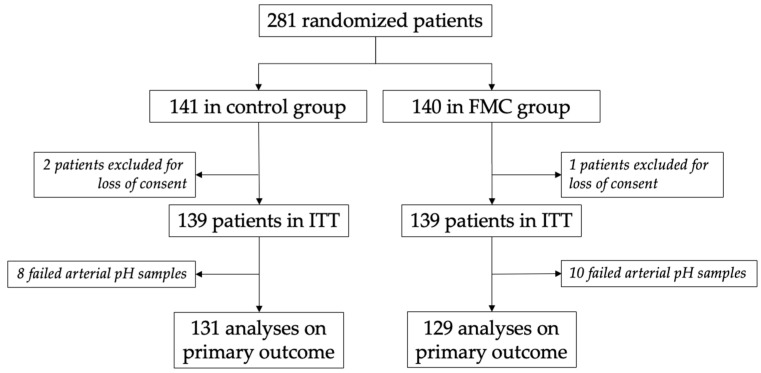
Flow chart. FMC: fetal movements counting; ITT: intention-to-treat.

**Figure 2 healthcare-10-02569-f002:**
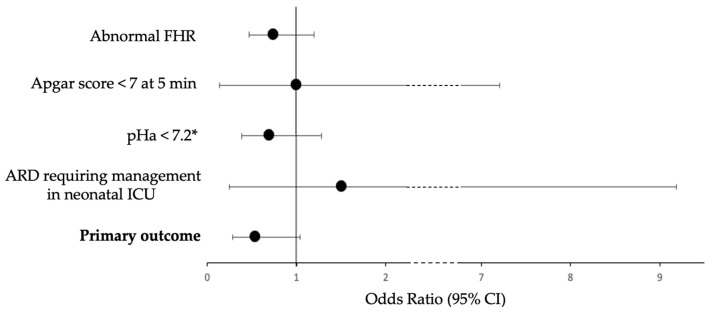
Odds ratio for neonatal morbidity criteria and their 95% confidence interval. * 18 missing data (10 from the “FM counting” group, 8 from the “control” group), analyses performed without missing data. FHR: fetal heart rate; pHa: umbilical cord arterial pH; ARD: acute respiratory distress; ICU: intensive care unit.

**Table 1 healthcare-10-02569-t001:** Demographic characteristics. Data are expressed as n (%) or means +/− standard deviations.

		FMC Group (n = 139)	Control Group (n = 139)	*p*-Value
Age (years)		31.10 +/− 4.62	30.68 +/− 4.59	0.45
	<35	107 (77.0)	114 (82.0)	0.30
	≥35	32 (23.0)	25 (18.0)	
BMI (Kg/m^2^)		25.14 +/− 5.56	25.00 +/− 5.48	0.83
	<20	14 (10.1)	22 (15.8)	0.35
	20–30	102 (73.4)	94 (67.6)	
	≥30	23 (16.6)	23 (16.6)	
Parity	Primiparas	64 (46.0)	67 (48.2)	0.72
	Multiparas	75 (55.0)	72 (51.8)	
Obstetrical history	Vaginal delivery	47 (33.8)	53 (38.1)	0.45
	Forceps	5 (3.6)	9 (6.5)	0.27
	Vacuum	12 (8.6)	6 (4.3)	0.14
	Cesarean section	10 (7.2)	9 (6.5)	0.81
	Miscarriage	26 (18.7)	23 (16.6)	0.64
	Ectopic pregnancy	2 (1.4)	3 (2.2)	0.65
	Legal abortion	10 (7.2)	14 (10.1)	0.39
	Therapeutic abortion	2 (1.4)	3 (2.2)	0.65
Smoking		13 (9.4)	12 (8.6)	0.83
	≥10 cigarettes/days	1 (7.7)	1 (8.3)	0.95
	Ex-smoker	10 (7.9)	14 (11.0)	0.40
Gestational diabetes		19 (13.7)	11 (7.9)	0.12

FMC: fetal movements counting; BMI: body mass index.

**Table 2 healthcare-10-02569-t002:** Neonatal outcomes. Data are expressed as n (%). Primary outcome is defined as the presence of at least two neonatal morbidity criteria.

	FMC Group (n = 139)	Control Group (n = 139)	OR (95% CI)	*p*-Value
Abnormal FHR	68 (48.9%)	78 (56.1%)	0.75 (0.47–1.20)	0.23
Apgar score of <7 at 5 min	2 (1.4%)	2 (1.4%)	1.00 (0.14–7.20)	1.00
pHa of <7.2 *	23 (17.8%)	31 (23.7%)	0.70 (0.38–1.28)	0.25
ARD requiring management in neonatal ICU	3 (2.2%)	2 (1.4%)	1.51 (0.25–9.18)	0.65
Primary outcome *	18 (14.0%)	30 (22.9%)	0.55 (0.29–1.04)	0.0630

* 18 missing data (10 from the “FM counting” group, 8 from the “control” group), analyses performed without missing data. FMC: fetal movement counting; FHR: fetal heart rate; pHa: umbilical cord arterial pH; ARD: acute respiratory distress; ICU: intensive care unit.

**Table 3 healthcare-10-02569-t003:** Outcomes related to consultations for decreased FM. Data are expressed as n (%).

		FMC Group (n = 139)	Control Group (n = 139)	*p*-Value
Consultation for decreased FM		8 (5.8)	7 (5.0)	0.79
	<12 h	3 (37.5)	5 (71.4)	0.31
	≥12 h	5 (62.5)	2 (28.6)	
Justified consultation for decreased FM		3 (37.5)	2 (28.6)	0.71
	Abnormal FHR	1 (12.5)	1 (14.3)	
	Manning	2 (25.0)	1 (14.3)	
	Kleihauer	0 (-)	0 (-)	

FMC: fetal movement counting; FM: fetal movement; FHR: fetal heart rate.

**Table 4 healthcare-10-02569-t004:** Outcomes related to obstetrical management. Data are expressed as n (%) or means +/− standard deviations.

		FMC Group (n = 139)	Control Group (n = 139)	*p*-Value
Mode of labor onset	Spontaneous	80 (57.6)	75 (54.0)	0.59
	Induction	57 (41.0)	61 (43.9)	
	Cesarean section before labor	2 (1.4)	3 (2.2)	
Mode of delivery	Spontaneous vaginal	91 (65.5)	77 (55.4)	0.23
	Operative delivery	29 (20.9)	38 (27.3)	
	Cesarean section	19 (13.7)	24 (17.3)	
Gestational age at delivery (days)		290.08 +/− 1.90	290.46 +/− 1.89	0.09
Indication for labor induction		n = 57	n = 61	
	Decreased FM	7 (12.3)	6 (9.8)	0.67
	Abnormal FHR	8 (14.0)	11 (18.0)	0.55
	41 weeks and 5 days	51 (89.5)	52 (82.3)	0.49
	Bishop score of >6	9 (15.8)	13 (21.3)	0.44
	PROM	9 (15.8)	4 (6.6)	0.11
	Others	2 (3.5)	9 (14.8)	
Labor induction method		n = 57	n = 61	
	Intravenous oxytocin	13 (22.8)	18 (29.5)	0.41
	Oral misoprostol	28 (49.1)	33 (54.1)	0.59
	Dinoprostone vaginal insertion	8 (14.0)	8 (13.1)	0.88
	Dinoprostone vaginal gel	2 (3.5)	0 (-)	0.23
	Cervical balloon	8 (14.0)	10 (16.4)	0.72
	Others	2 (3.5)	9 (14.8)	
Indication for operative delivery		n = 29	n = 38	
	Abnormal FHR	14 (48.3)	27 (71.1)	0.058
	Inefficient expulsive effort	19 (65.5)	17 (44.7)	0.16
	Posterior presentation	3 (10.3)	1 (2.6)	0.31
Indication for cesarean section		n = 19	n = 24	
	Abnormal FHR	10 (52.6)	11 (45.8)	0.66
	Failure of operative delivery	3 (15.8)	3 (12.5)	0.75
	Stop of cervix dilatation	7 (36.8)	12 (50.0)	0.39
	Others	6 (31.6)	5 (20.8)	

FMC: fetal movement counting; FM: fetal movement; FHR: fetal heart rate; PROM: pre-labor rupture of membranes.

**Table 5 healthcare-10-02569-t005:** Factors influencing the primary outcome. Data are expressed as n (%) or means +/− standard deviations.

		No Morbidity (n = 212)	Neonatal Morbidity (n = 48)	*p*-Value
Age (years)		30.93 +/− 4.63	30.33 +/− 4.54	0.42
	<35	169 (79.7)	39 (81.3)	0.81
	≥35	43 (20.3)	9 (17.3)	
BMI (Kg/m^2^)		25.21 +/− 5.50	24.28 +/− 5.39	0.29
	<20	22 (10.4)	10 (20.8)	0.12
	20–30	152 (71.7)	32 (66.7)	
	≥30	38 (17.9)	6 (12.5)	
Parity	Primiparas	92 (43.4)	29 (60.4)	0.03
	Multiparas	120 (56.6)	19 (39.6)	
Obstetrical history	Vaginal delivery	80 (31.7)	13 (27.1)	0.16
	Forceps	11 (5.2)	2 (4.2)	0.77
	Vacuum	15 (7.1)	1 (2.1)	0.32
	Cesarean section	17 (8.0)	2 (4.2)	0.54
	Miscarriage	36 (17.0)	9 (218.8)	0.77
	Ectopic pregnancy	3 (1.4)	1 (2.1)	0.56
	Legal abortion	21 (9.9)	2 (4.2)	0.27
	Therapeutic abortion	4 (1.9)	0 (-)	0.34
Smoking		18 (8.5)	6 (12.5)	0.41
Gestational diabetes		24 (11.3)	5 (10.4)	0.86
Consultation for decreased FM		10 (4.7)	4 (8.3)	0.32
	<12 h	4 (40.0)	3 (75.0)	0.56
	≥12 h	6 (60.0)	1 (25.0)	
Justified consultation for decreased FM		2 (20.0)	2 (50.0)	0.52
	Abnormal FHR	0 (-)	2 (50.0)	
	Manning	2 (20.0)	0 (-)	
	Kleihauer	0 (-)	0 (-)	
Mode of labor onset	Spontaneous	128 (61.5)	19 (39.6)	<0.01
	Induction	80 (38.5)	29 (60.4)	
Mode of delivery	Spontaneous vaginal	140 (66.0)	17 (35.4)	<0.001
	Operative delivery	42 (18.8)	21 (43.8)	
	Cesarean section	30 (14.2)	10 (20.8)	
Gestational age at delivery (days)		290.11 +/− 1.86	290.81 +/− 1.84	0.02
Indication for labor induction		n = 80	n = 29	
	Decreased FM	8 (10.0)	4 (13.8)	0.73
	Abnormal FHR	13 (16.3)	5 (17.2)	0.90
	41 weeks and 5 days	68 (85.0)	26 (89.7)	0.75
	Bishop score of >6	16 (20.0)	4 (13.8)	0.46
	PROM	9 (11.3)	3 (10.3)	0.89
Labor induction method		n = 80	n = 29	
	Intravenous oxytocin	24 (30.0)	5 (17.2)	0.18
	Oral misoprostol	35 (43.8)	19 (65.6)	0.04
	Dinoprostone vaginal insertion	11 (13.8)	5 (17.2)	0.76
	Dinoprostone vaginal gel	2 (2.5)	0 (-)	0.39
	Cervical balloon	16 (20.0)	2 (6.9)	0.15
Indication for operative delivery		n = 42	n = 21	
	Abnormal FHR	20 (47.6)	18 (85.7)	<0.01
	Inefficient expulsive effort	29 (69.0)	5 (23.8)	<0.001
	Posterior presentation	3 (7.1)	1 (4.8)	0.71
Indication for cesarean section		n = 30	n = 10	
	Abnormal FHR	14 (46.7)	7 (70.0)	0.28
	Failure of operative delivery	4 (13.3)	1 (10.0)	0.78
	Stop of cervix dilatation	15 (50.0)	3 (30.0)	0.46

BMI: body mass index; FM: fetal movement; FHR: fetal heart rate; PROM: pre-labor rupture of membrane.

## Data Availability

The data presented in this study are available on request from the corresponding author.

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
