# Peer review of "Fetal Movement Counting in Prolonged Pregnancies: The COMPTAMAF Prospective Randomized Trial"

_healthcare, 2022, doi:10.3390/healthcare10122569_

Round 1

Reviewer 1 Report

This manuscript corresponds to a prospective single center randomized open-label study aiming to evaluate whether fetal mouvements counting could improve neonatal outcomes in prolonged pregnancies. 

Introduction: no comment

Material and methods section:

It is not clear why patients were induced at 41+5 GW rather than 41+6 or 42+0 GW? 

What about in utero fetal demise? None was observed in this study but one expected to read this terrible outcome included in the primary outcome

It would be nice to add as supplemental material the french written flyer given to explain how to count fetal movements and how to register them (chart)

Results section:

Neonatal morbidity expressed as the composite score was not statistically different between the 2 groups but a trend for fewer events in the "FM counting" group was observed. I suggest to emphasize this aspect. 

Table 5: why not giving one single p value concerning "indication for operative delivery" and "indication for cesarean section"?

Discussion: Good

line 212: 40,000 to 40,000 patients (mistake in numbers)

line 219: semester instead of trimester

Conclusion:

I suggest to write "Fetal movement counting for women in prolonged pregnancy failed to demonstrated a significant reduction in neonatal morbidity" rather than "is not associated with"

Author Response

Dear Reviewer,

Thank you for the review of our manuscript “Fetal movements counting in prolonged pregnancies: a pro-spective randomized trial COMPTAMAF” (manuscript ID healthcare-2068025). Please find herewith the revised version, based on the reviewers’ comments.

This manuscript corresponds to a prospective single center randomized open-label study aiming to evaluate whether fetal mouvements counting could improve neonatal outcomes in prolonged pregnancies.

Introduction: no comment

Material and methods section:

It is not clear why patients were induced at 41+5 GW rather than 41+6 or 42+0 GW?

We clarified why patients were induced at 41+5 GW (because induction can take up to 48 hours, and neonatal risks are much more substantial after 42 GW).

What about in utero fetal demise? None was observed in this study but one expected to read this terrible outcome included in the primary outcome.

We also specified that a case of in utero demise, would have been considered as an occurrence of the primary outcome.

It would be nice to add as supplemental material the french written flyer given to explain how to count fetal movements and how to register them (chart)

We add as supplemental material the original flyers given to each group to explain the fetal movements and the chart to register them.

Results section:

Neonatal morbidity expressed as the composite score was not statistically different between the 2 groups but a trend for fewer events in the "FM counting" group was observed. I suggest to emphasize this aspect.

We changed the formulation of the results of primary outcomes, to emphasize the trend of fewer events in the “FM counting” group.

Table 5: why not giving one single p value concerning "indication for operative delivery" and "indication for cesarean section"?

We underwent statistical analyses for all the data (indication for operative delivery or for C-section) comparing to all the operative delivery or C-section for other indications. Thus, we can see for every indication if there is or not more neonatal morbidity due to the specific indication.

Discussion: Good

line 212: 40,000 to 40,000 patients (mistake in numbers)

line 219: semester instead of trimester

We have made the corrections

Conclusion:

I suggest to write "Fetal movement counting for women in prolonged pregnancy failed to demonstrated a significant reduction in neonatal morbidity" rather than "is not associated with"

We also changed the conclusion, according to suggestions made by the two reviewers.

Reviewer 2 Report

Nice paper. Some formal comments :

In the abstract :

Numbers should be spelled out after a full stop.

It should be noted that in this study the counting of fetal movements does not affect neonatal outcomes.

Discussion :

Apgar score and not APGAR score.

Conclusion:

It is necessary to correct : “fetal movement counting in prolonged pregnancies is not associated 266 with a reduction of adverse neonatal outcomes”

References:

Journals are cited in full or in abbreviated format... Submission rules should be followed.

Author Response

Dear Reviewer,

Thank you for the review of our manuscript “Fetal movements counting in prolonged pregnancies: a pro-spective randomized trial COMPTAMAF” (manuscript ID healthcare-2068025). Please find herewith the revised version, based on the reviewers’ comments.

Nice paper. Some formal comments :

In the abstract :

Numbers should be spelled out after a full stop.

We edited formulation to avoid numbers after full stop

It should be noted that in this study the counting of fetal movements does not affect neonatal outcomes.

Discussion :

Apgar score and not APGAR score.

 We have made the corrections

Conclusion:

It is necessary to correct : “fetal movement counting in prolonged pregnancies is not associated 266 with a reduction of adverse neonatal outcomes”

We also changed the conclusion, according to suggestions made by the two reviewers.

References:

Journals are cited in full or in abbreviated format... Submission rules should be followed.

We have corrected journal citation in abbreviation format, according to submission rules.

Thank you for your consideration of this manuscript.

Sincerely,

Louise Moniod